# Local seed sourcing for sustainable forestry

**Ricardo Alía[1]\*, Eduardo Notivol[2], José Climent[1], Felipe Pérez[3], Diana Barba[1], Juan Majada[4], José Manuel García del Barrio[1]**

**1** Department of Ecology and Forest Genetics, Institute of Forest Sciences, INIA-CSIC, Madrid, Spain,
**2** Department of Environment, Agricultural and Forest Systems, CITA, Zaragoza, Spain, **3** Directorate General of Biodiversity, Forest and Desertification, MITECO, Madrid, Spain, **4** CETEMAS, Carbayin, Spain

\* alia@inia.csic.es

**Data Availability Statement:** Data are available from the Zenodo repository (DOI: 10.5281/zenodo. 7157589).

**Funding:** This research was supported by the Project AEG 17-048 established in the frame of the measure 15.2 "support to the conservation and use

## Abstract

Seed sourcing strategies are the basis for identifying genetic material meeting the requirements of future climatic conditions and social demands. Specifically, local seed sourcing has been extensively promoted, based on the expected adaptation of the populations to local conditions, but there are some limitations for the application. We analyzed Strict-sense local and Wide-sense local (based on climatic similarity) seed sourcing strategies. We determined species and genetic pools based on these strategies for 40 species and deployment zones in Spain. We also obtained the total number of seed sources and stands for these species in the EU countries. We analyzed the richness of the pools, the relationship with variables related to the use of the species in afforestation, and the availability of seed production areas approved for the production of reproductive material destined to be marketed. This study confirms the existence of extensive species and genetic local pools. Also, that the importance of these pools differs for different species, limitations being derived from the use of forest reproductive material and the existence of approved basic materials. Strategies derived from local seed sourcing approaches are the basis for the use of forest reproductive material because a large number of the species in the area considered in the study are under regulation. However, despite the extensive work done to approve basic materials, limitations based on the availability of seed production areas to provide local material for sustainable forestry are found in those species. Considering a Wide-sense local seed sourcing strategy we provide alternative pools in order to meet social demands under the actual regulations on marketing of reproductive materials.

## Introduction

Sustainable forest management aims at maintaining the biodiversity, productivity, regeneration capacity, vitality and potential of forests (Resolution H1, Forest Europe 1993), taking into account the economic value of the ecosystem services they provide [1], and the urgent need to increase forest resilience [2]. In this context, both artificial and natural regeneration play essential roles in ensuring resilience [3], long-term population persistence [4] and the restoration of ecosystem functionality [5, 6].

of forest genetic resources" and under Regulation (EU) No 1305/2013 of the European Parliament and of the Council of 17 December 2013 on support for rural development by the European Agricultural Fund for Rural Development (EAFRD) with 75% co-financing (FP). This study was funded by the European Union Horizon 2020 research and innovation programme under grant agreement No 773383 (B4EST project) (RA), and the Ministry of Science (RTI2018-094691-B-C32) (JC). The funders had no role in study design, data collection and analysis, decision to publish, or preparation of the manuscript.

**Competing interests:** The authors have declared that no competing interests exist.

While natural regeneration is always based on local genetic resources, artificial regeneration often makes use of the deliberate transfer of genetic resources from elsewhere. In the last case, seed sourcing involves matching the reproductive material collection area (Seed Production Area (SPA, [7]), region of provenance or seed zone) and the area where the material will be used (deployment zone) [8].

The origin of the planting stock is a major concern in forestry since it influences (as shown by extensive provenance research [9]) the existing genetic resources [10] and the future performance and adaptability [11] of populations. The election of a local provenance (*local seed sourcing*) is based on the expectation that populations are locally adapted [12–14], as origin also influences different traits related to the stability, adaptation, resistance, productivity and diversity of the planting stock [14, 15]. Local seed sourcing is the preferred, most widespread approach [16] in biodiversity conservation [17]. It is recommended as a general principle in Forest Europe and is followed by many countries [3]. It can be interpreted as *strict-sense* when the material used in the deployment location was also collected there, or as *wide-sense* when the material used in the deployment location was collected from locations with similar climate [16, 18].

Other strategies have been suggested under climate change scenarios, such as climate-predictive, composite, admixture, climate-adjusted, and assisted migration [16, 19], which require information on the performance of the planting stock under different environments [20–27]. A given seed-sourcing strategy makes it possible to define a species pool, i.e. the set of species that can potentially inhabit a site due to suitable local ecological conditions [28], and a genetic pool within those species [29], i.e. the set of genetic materials suitable for the site.

Although these pools can be defined only theoretically, their use for specific purposes (e.g., for forestry), requires that SPA should meet the regulations (*e.g.* European Union, OCDE, USA schemes [30–32]) to be approved for the production of reproductive materials (i.e. seeds, fruits, plants or part of plants). These regulations define different basic materials from which they can be obtained (e.g. seed-sources, stands, seed orchards, parents of families, clones and mixture of clones) and different categories of reproductive materials (e.g. source-identified, selected, qualified and tested). They also define basic units for the marketing of source-identified and selected categories of reproductive materials (regions of provenance or seed zones, depending on the scheme). Among all the types of basic materials, SPA for producing source-identified and selected reproductive materials presents some advantages in the context of sustainable forestry. They are collected from populations of known origin, high population size and no phenotypic selected or selected stands. They are also the most frequent of all the existing basic materials (in the EU, for instance, they represent 79.2% of the total basic materials). However, the amount of species under regulation, the presence of different areas for afforestation, and the cost of collecting reproductive material from many different zones, haves shown a reduced availability that amount a major limitation in forestry and ecosystem restoration [33, 34].

We used Peninsular Spain and the Balearic Islands as our model study area, and expanded our study to SPA (basic materials for producing source-identified and selected reproductive materials) in the EU. Our model study area presents a higher tree diversity compared to other areas in Europe [35], as it is a well-known biodiversity hot-spot with a strong interest in species conservation [36]. In this area, there is a long tradition of multi-purpose forest plantations with different goals [37]: restoration, protection and, to a lesser extent, production [38]. There is a preference for source-identified and selected reproductive materials, from a highly diverse species pool [39], that represent 97% of the total reproductive material produced annually in Spain. The other two categories (qualified and tested materials) represent 3% of the basic materials produced for all the species.

We used the defined regions of provenance [40] and deployment zones [41] to evaluate two seed-sourcing strategies: Strict-sense local and Wide-sense local, taking into account the landscape level, and considering the reported scale of gene flow in forest tree species [42]. To do so, we analysed 40 native forest tree species with actual trading of reproductive materials in Spain, and identified the regions of provenance that were climatically suitable for each species and deployment zone. We also defined the availability of SPA. Our objectives were to determine the richness of the pools for each of the two seed-sourcing strategies, per species and deployment zone, and the relationship among richness, the use of the species reproductive material and the climatic variables of the deployment zones. To conclude, we further discuss the application of local seed-sourcing for afforestation, restoration and reforestation, in a highly diverse environment such as the Mediterranean forests, and its application in other areas in Europe in a context of sustainable forestry.

## Material and methods

### Species and pools

We considered 40 species (Table 1) regulated for marketing and trading of forest reproductive material in Spain. They were classified depending on whether that material was mainly used for restoration, for protection and to a lesser extent, in productive plantations according to the published national guidelines [38].

**Table 1. Species considered in the study.**

| Code[1] | Species | Use[2] | Regpro[3] | Code | Species | Use | Regpro |
|---|---|---|---|---|---|---|---|
| Aal | *Abies alba* Mill. | RE | 6[a] | psy | *Pinus sylvestris* L. | PR,PT | 19[a] |
| api | *Abies pinsapo* Boiss. | RE | 3[a] | pun | *Pinus uncinata* Ram. ex DC. | PT | 6[a] |
| apl | *Acer platanoides* L. | RE | 7[d] | pav | *Prunus avium* L. | PR,RE | 33[d] |
| aps | *Acer pseudoplatanus* L. | RE | 19[d] | qca | *Quercus canariensis* Willd. | RE | 5[a] |
| aun | *Arbutus unedo* L. | RE | 47[d] | qco | *Quercus coccifera* L. | RE | 33[d] |
| bpu | *Betula pubescens* Ehrh. | RE | 23[d] | qfa | *Quercus faginea* Lam. | RE | 27[a] |
| csa | *Castanea sativa* Mill. | PR | 37[d] | qil | *Quercus ilex* L. | PR,RE | 28[a] |
| fsy | *Fagus sylvatica* L. | PT | 18[a] | qpe | *Quercus petraea* (Matt.) Liebl. | RE | 13[a] |
| fex | *Fraxinus excelsior* L. | RE | 17[d] | qpu | *Quercus pubescens* Willd. | RE | 6[a] |
| iaq | *Ilex aquifolium* L. | RE | 30[d] | qpy | *Quercus pyrenaica* Willd. | RE | 28[a] |
| jre | *Juglans regia* L. | PR | 39[d] | qro | *Quercus robur* L. | RE | 11[a] |
| jco | *Juniperus communis* L. | RE | 31[d] | qsu | *Quercus suber* L. | PR,RE | 25[a] |
| jox | *Juniperus oxycedrus* L. | RE | 45[d] | sar | *Sorbus aria* (L.) Crantz | RE | 31[d] |
| jph | *Juniperus phoenicea* L. | RE | 36[d] | sau | *Sorbus aucuparia* L. | RE | 22[d] |
| jth | *Juniperus thurifera* L. | PT | 28[d] | tga | *Tamarix gallica* L. | RE | 28[d] |
| oeu | *Olea europea* Brot. | RE | 45[d] | tba | *Taxus baccata* L. | RE | 26[d] |
| pha | *Pinus halepensis* Mill. | PT | 20[a] | tco | *Tilia cordata* Mill. | RE | 14[d] |
| pni | *Pinus nigra* Arn. | PR,PT | 15[a] | tpl | *Tilia platyphyllos* Scop. | RE | 18[d] |
| ppa | *Pinus pinaster* Aiton. | PR,PT | 29[a] | ugl | *Ulmus glabra* Huds. | RE | 21[d] |
| ppe | *Pinus pinea* L. | PR,PT | 12[a] | umi | *Ulmus minor* Mill. s.l. | RE | 46[d] |

[1]**Code**: According to the Commission Regulation (EC) No 1597/2002 of 6 September 2002.

[2]**Use:** from Peman et al. (2013). RE: restauration, PR: production, PT: protection.

[3]**Regpro**: number of regions of provenance of the species

([a]Agglomerative method,

[d]Divisive method).

Distribution data were available for each species from the Spanish National Inventory and Spanish Forest Map (1/25000) and transformed into species presence/absence data in a grid of 1x1 km$^2$. We considered natural populations–i.e., excluding plantations- in the Spanish Iberian Peninsula and Balearic Islands, excluding plantations, following each species' regions of provenance [40].

*Deployment zones* were established by a division of Spain into continuous regions with similar environmental characteristics and following a biogeographical classification [43], resulting in fifty units [41, 44] (S1 Fig).

*Regions of provenance*, are the basic marketing units for source- identified and selected materials according to EU regulations [30]. The regions were defined for all the species considered in this study (Table 1) using two different methodologies: agglomerative and divisive [40]. The agglomerative method (see [45]), groups each species populations with similar ecological, phenotypic or genetic characteristics. These species-specific regions of provenance were defined for 17 main forest species: *Abies* (2), *Fagus* (1), *Pinus* (6) and *Quercus* (8). The divisive method splits the territory into continuous regions with similar environmental characteristics using a biogeographical classification [41]. This method was applied for the remaining 23 species in the in the study and resulted in regions that coincided with the deployment zones and were not species-specific.

**Seed sourcing strategies.** A *Strict-sense local seed-sourcing strategy* (SSL) was defined for each deployment zone as the use for each species of autochthonous populations pertaining to the same region of provenance. *Wide-sense local seed-sourcing strategy* (WSL) was defined as the use, in each zone, of regions of provenance with a suitable climatic niche.

Niche modelling was used to obtain the climate-predicted *distribution* of the species based on actual presence and climate (assuming no restrictions to dispersal and no human influence) (see [39] and S1 Annex). Eight climatic variables already used for the niche modelling of different species in Spain [46] were used: Rainfall (P, mm), summer precipitation (SP, mm), winter precipitation (WP), Dry period, considered when P<2T (DP, months), Frost period, considered when T<0˚C (FP, months), Annual mean temperature (TM, Celsius degrees), Mean of the Maximum temperatures of the hottest month (MXHM, Celsius degrees) and Mean of the minimum temperatures of the coldest month (*MNCM*, Celsius degrees). Climatic data was obtained for a 1 km$^2$ square grid. A similarity index [47] was obtained based on the Mahalanobis climatic distance (using the same set of climatic variables as in the niche modelling estimation, see S1 Annex), comparing the points present both in the region of provenances and each deployment zone. Based on this similarity, the regions of provenance were classified as suitable or not for each deployment zone.

A dataset was created including the regions of provenance identified for each of the 40 species and 50 deployment zones using the SSL and WSL strategies (DOI: 10.5281/zenodo.7157589).

**Seed sourcing pools.** We defined eight seed-sourcing pools from this database by combining the two seed-sourcing strategies SSL and WSL at two levels (species and genetic) and considering the availability of SPA in the national register.

The *species pool* was defined as the set of species for each deployment zone following each strategy. Similarly, the *genetic pool* was defined as the set of regions of provenance defined by each strategy.

The *available pool* was defined as those regions of provenance with at least one SPA in the national register (source-identified or selected categories). The National Register held by the National Authority includes the existing approved basic materials (https://www.miteco.gob.es/es/biodiversidad/temas/recursos-geneticos/geneticos-forestales/rgf_catalogo_materiales_base.aspx; data accessed 31/12/2021).

**Data on reproductive material.** The actual use of the species in afforestation and reforestation programs was defined by the following variables: Number of region of provenances of the species (*regpro*); Mean number of SPA (entries or accessions in the national register) by region of provenance (*spa*); Afforested area/year in ha (*aff_su*); Ratio of public afforestation to total afforested area (*aff_pu*); Ratio afforested area for protection to total afforested area (*aff_pr*); Source-identified and selected reproductive material by year in number of plants (*frm_si*) and Qualified and tested reproductive material by year in number of plants (*frm_qt*). Data on afforestation and production of reproductive material were obtained as the mean for the period 2005–2016 (last year with available data) using on the Spanish forestry statistics.

We also calculated the percentage of deployment zones with endangered local populations (*recgen*) for each species, following the criteria in the Spanish Strategy for the conservation and sustainable use of forest genetic resources [40, 48]. This value was used as a proxy to the genetic risk of transferring materials to a given zone.

**Pool of basic material by country in the EU.** We computed the available pool of basic material of the species in our study compiled for each EU country. The FOREMATIS database (https://ec.europa.eu/forematis/) includes the location of the approved basic material, type, origin and purpose by species and country (Community List of Approved Basic Material for the Production of Forest Reproductive Material [48]). We obtained the mean and the harmonic mean of the basic material for all the species in each country (equivalent to an effective number per species) for two classes: SPA (seed sources and stands) and improved basic for qualified and tested categories (seed orchards, clones, parent of families).

## Statistical analysis

**Richness of seed sourcing pools per deployment zone.** We obtained the richness of the eight pools (already defined for each deployment zone) and analyzed the climatic variables related to the richness increment between Strict-sense and Wide-sense local for the different categories (species, genetic and available genetic). A stepwise linear selection model was applied (lm function, stepAIC option in R), to select the variables explaining the relationship among richness increment pools and climatic variables. The procedure start with a saturated model, and the least significant variables are removed until no further decrease in the Bayesian Information Criterion (BIC). Non-significant variables ($\alpha > 0.05$) were also removed from the final selected model. Mean values of the climatic variables used in niche modelling, and the altitude for each deployment zone, were computed based on the 1 km grid, and standardized by the mean and standard deviation across deployment zones.

**Relationship among seed sourcing pools and use of the species.** We explored the relationship among the set of variables describing the seed sourcing pools by species (richness of the genetic and available genetic pools, number and harmonic mean of deployment zones -equivalent to the effective number of deployment zones- according to the frequency of the species' present in the region), and the variable set describing the actual deployment of reproductive materials (see above). A canonical correspondence analysis (CCA) [49], was performed and the biplot of variables and species was used to explore the relationship among these two sets of variables.

Additionally, we explored the relationship among the variables related to the seed sourcing pools and the variables describing the use of the species for two main type of species depending on the method–agglomerative of divisive- for establishing the regions of provenance. A T-test for two means with unknown population standard deviations (t.test function in the stat package in R), was computed from the means and standard deviation of each of the two groups of species.

**Basic materials in the EU.** We compared by ANOVA (lm function in R) the values of the number of species under regulation in each country from the 40 considered in the study, and the mean values for the SPA and improved basic materials for different EU regions (Nordic, Western Central, Eastern Central, Western South, Eastern South).

All statistical analyses were made within the R environment (R Core Team, 2015), with the packages corrplot [50], Hmisc [51], CCA [49], and MASS [52].

## Results

### Seed sourcing pools by deployment zone

The species differed in their range of distribution, and therefore in the local materials available for each deployment zone. The richness of the species pool (for both SSL and WSL strategies) varied greatly among deployment zones. The mean value of the SSL species pool was 20.2 ± 7.6 (range from 4 to 36 species) with a slight increment for the WSL species pool to 26.9 ± 8.1 (range from 8 to 39) (Fig 1 and S1 Table).

When considering the genetic pools, richness increased greatly for both seed-sourcing strategies: to 27.7 ± 12.6 (range from 4 to 63) for SSL and to 114.1±57.8 (range from 10 to 245) for WSL. However, the available genetic pool, (SPAs in the national register), for all species, decreased to 17.7 ± 10.1 (60.2% of the total) for SSL and 57.6 ± 33.9 (48.4% of the total) for WSL strategy.

The increment of richness among SSL and WSL pools showed a climatic trend mainly associated to temperature for the three levels considered: species, genetic and available genetic (Table 2), with a reduction in the areas with higher temperatures, in southern Spain.

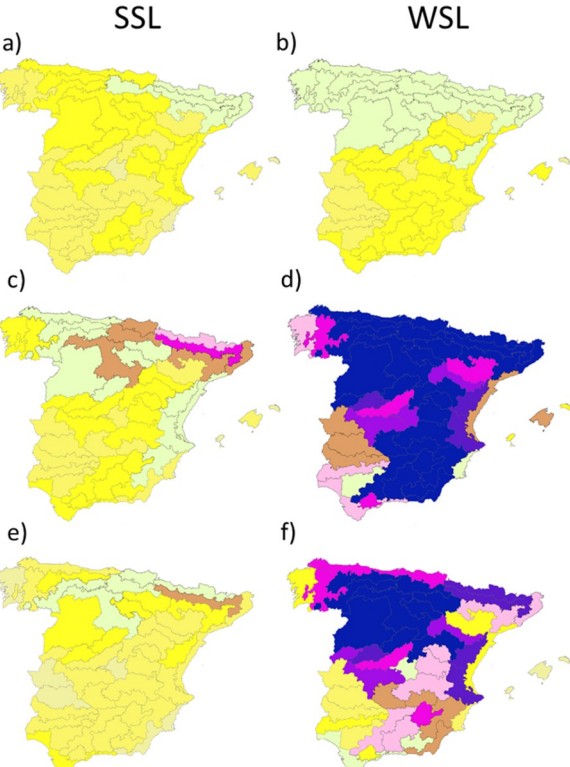

**Fig 1.** Richness of pools for Strict-sense and Wide-sense local seed-sourcing strategies for the species (a, b), genetic (c, d) and available genetic pools (e, f). BDLJE CC-BY 4.0 ign.es, DR miteco.gob.es.

## Seed sourcing genetic pools per species

The richness per deployment zone of genetic pools differed greatly between SSL and WSL strategies: 1.5±0.7 and 4.0±1.6, respectively. Moreover, the available pools were significantly lower (1.0±0.8 and 2.1±0.9 respectively). For the 40 species, only 41% of the regions of provenance had at least one basic material (source-identified or selected categories) in the national register, with a mean value of 5.8 basic materials per region of provenance (S2 Table).

Nevertheless, the pools can be used in a significantly higher number of deployment zones (133% as an average increment, from 25.3±12.6 in the Strict-sense local to 33.6±14.1 in the Wide-sense local). This value is reduced when considering availability (15.1±10.0 and 26.2 ±14.1 respectively).

The canonical correlation analysis showed a trend in the biplot of species on the two first components based on variables related to the use of forest reproductive material, with a transition from those species more extensively used to those used mostly for ecological restoration (Fig 2 and S3 Table). The species with an agglomerative method have a lower number of regions of provenance, a higher amount of SPAs, higher afforested surface, and a higher production of forest reproductive material, including also a higher number of endangered populations (S4 Table).

When comparing the SPA pool richness in Europe, there were large differences among countries (Figs 3 and 4), with a higher number of species and higher number of SPA in southern countries, but the variability among countries within a region was still quite high.

## Discussion

We compared the effects of the potential genetic diversity (proxied by the number of suitable and available basic materials) of two alternative local seed sourcing strategies for 40 species in Spain.

**Table 2. Regression analysis for the increment of richness for different seed sourcing pools (strict-sense local and Wide-sense local) per deployment zone and climatic variables.** Variables retained after a stepwise selection model starting with the eight standardized variables.

| | Estimate | Std. Error | t-value | Pr(>\|t\|) | Analysis of variance | | |
| --- | --- | --- | --- | --- | --- | --- | --- |
| | | | | | M. Squares | F-value | Pr(>F) |
| *Species pool* | | | | | | | |
| ALT | -23.642 | 0.9222 | -2.564 | 0.014025 | 31.276 | 7.1488 | 0.010640 |
| P | 19.680 | 0.5541 | 3.551 | 0.000961 | 42.224 | 9.6509 | 0.003387 |
| MXHM | -39.996 | 0.7162 | -5.585 | 1.57e-06 | 42.865 | 9.7974 | 0.003175 |
| MNCM | -59.040 | 11.896 | -4.963 | 1.20e-05 | 107.765 | 24.6316 | 1.2e-05 |
| *Genetic pool* | | | | | | | |
| P | -14.999 | 7.379 | -2.033 | 0.04801 | 1080 | 0.9254 | 0.341203 |
| SP | -17.473 | 9.397 | -1.859 | 0.06952 | 12337 | 10.5714 | 0.002179 |
| TM | -76.989 | 11.677 | -6.593 | 4.09e-08 | 47407 | 40.6211 | 8.696e-08 |
| FRO | -29.741 | 9.189 | -3.237 | 0.00227 | 12225 | 10.4754 | 0.002273 |
| *Available Genetic pool* | | | | | | | |
| P | -9.846 | 3.645 | -2.702 | 0.00976 | 1671.1 | 6.3874 | 0.015161 |
| TM | -67.208 | 21.459 | -3.132 | 0.00309 | 17157.3 | 65.5788 | 2.918e-10 |
| FRO | -11.473 | 4.402 | -2.606 | 0.01245 | 2188.7 | 8.3658 | 0.005923 |

ALT: Altitude, P: rainfall, TM: Mean annual Temperature, SP: Summer Precipitation, FRO: Frost period, MXHM: Maximum temperature of the hottest month; MNCM: Minimum temperature of the coldest month. Arid Period was not retained for any case

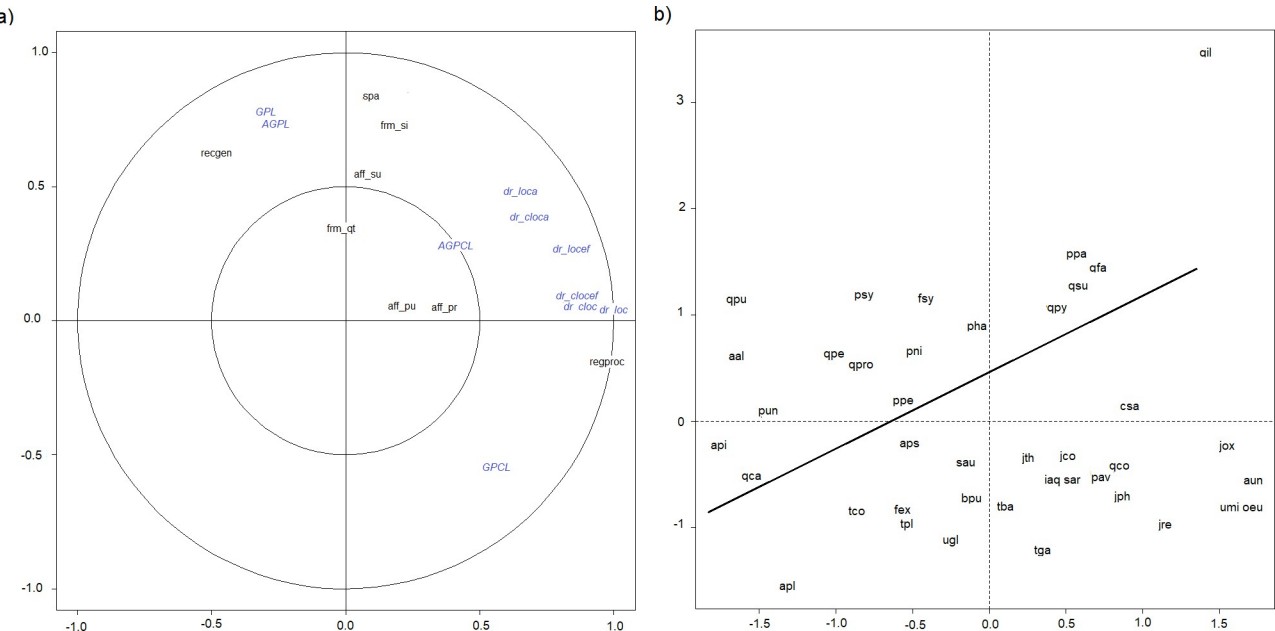

**Fig 2. Biplot of the two first components of a CCA analysis on variables related to the seed sourcing pools and variables related to the use of reproductive material for each of the 40 studied species.**

Our analysis is based on deployment zones and regions of provenance in contrast with other studies that work at a finer scale, i.e. the population level (e.g. [26, 53]), because we need to consider the scale at which genetic variation has implications in the use of genetic resources. Regions of provenance are defined using different criteria among and within countries [54], usually within genetic groups defined by neutral markers [55–57]. These regions are in the same order of magnitude as deployment zones. For autochthonous or indigenous populations (if the origin is known), region of provenance can be considered as that of populations within gene-flow distance and therefore suitable for local seed-sourcing. This is in agreement with the results on the response curves of different species which show a range where the populations are close to an optimum [13]. Accordingly, we can assume that the scale at which we define the separate genepools and the deployment zones are adequate for sustainable forestry and restoration activities.

We found that the Strict-sense local species pool was quite high in most of the deployment zones (mean value of 22), with wide genetic pools for those species. We assumed that climate prediction is a good estimate for the performance of the material when we considered Wide-sense local seed sourcing in our study, that is, that the material from more ecologically similar procurement zones would be preferred for a given deployment zone [see 58,59]. This is a more generalized local seed sourcing method than can be implemented easily for many different species. By using this strategy, the number of basic materials expected to be adapted to the local conditions increased from an average of 1.4 per species and deployment zone to 2.3. This strategy is still rather conservative, discarding recommendations of climatically distant basic materials for a given deployment zone.

We found an expectable climatic trend in the richness increment of the pools, associated with higher annual rainfall (or reduction of drought period) and temperatures. This is a general trend for species diversity, where annual precipitation and mean annual temperature play key favorable roles [60]. The species pool seems large enough for most of the goals of the

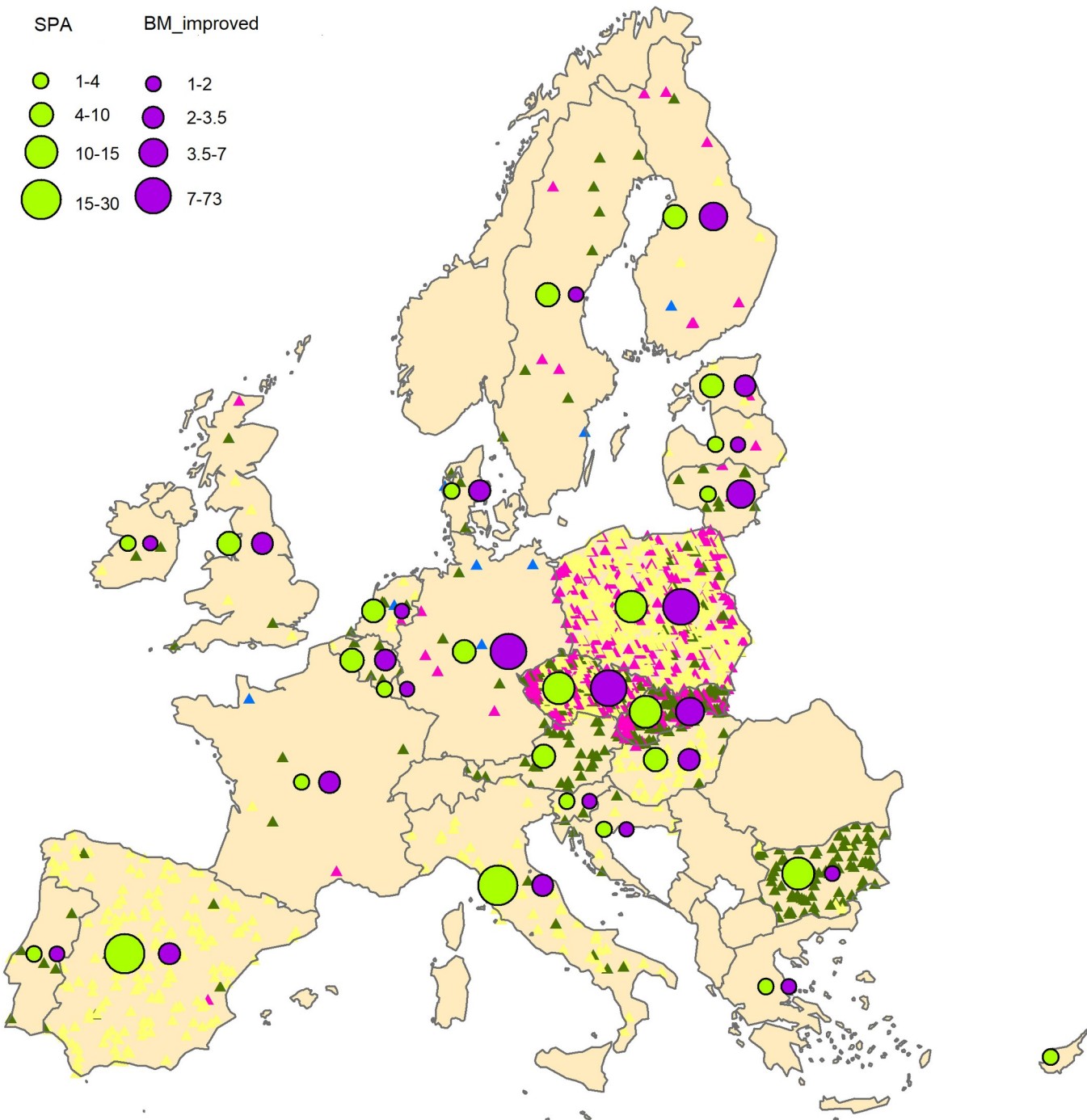

**Fig 3. Basic material in the EU for the 40 analyzed species.** SPA: harmonic mean of number of seed production areas by country; BM_improved: harmonic mean of number of basic material for qualified and tested categories. Color triangles: Proportional representation of different source categories of forest reproduction material (yellow triangles are identified, green ones are selected, pink ones are qualified and blue ones are tested). Source of basic material: www.forematis.eu. Political map of Europe: https://es.m.wikipedia.org/wiki/Archivo:Political_Map_of_Europe-en.svg. CC BY 4.0.

plantations (restoration, revegetation, afforestation), allowing the establishment of mixed forests with species differing in functional traits [61] in order to obtain multifunctional forests [62], something particularly relevant in the Mediterranean region.

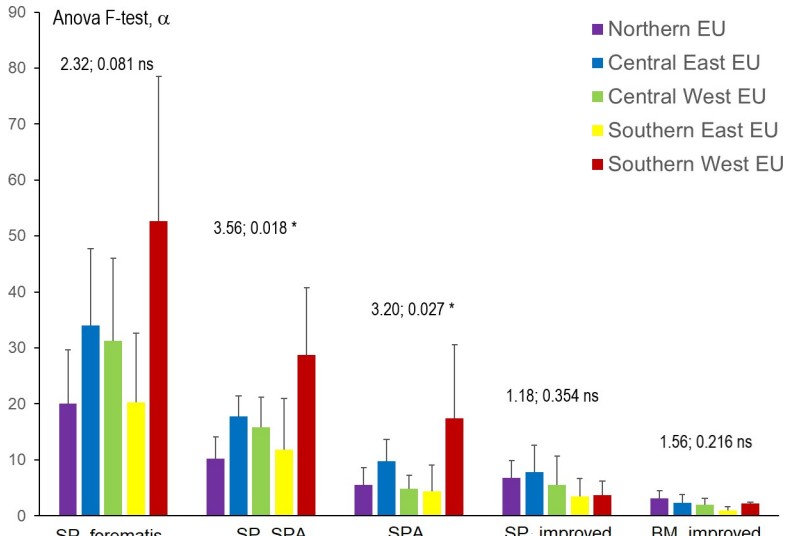

**Fig 4. Comparison of basic material in EU regions.** SP_Forematis: number of species for each country with approved basic materials; SP_SPA: number of species for each country with basic materials included in this study; SPA: harmonic mean of number of SPA; SP_improved: number of species for each country with approved basic material for the production of qualified and tested reproductive material (excluding clones and parent of families), BM_improved: harmonic mean of number of basic material for the production of qualified and tested reproductive material. Anova F-test and significance level (α) is included for each of the variables analyzed. Source www.forematis.eu. CC-BY 4.0.

Despite the widespread use of Strict-sense local seed sourcing in many countries and regions, there are relevant concerns about this strategy. Local is not always the best option, for different reasons: the absence of native forest species in the deployment zone is a first obvious one. Moreover, low performance or bad adaptation of the local material has also been reported in some cases [63–66], deriving from reduced population size [67], poor seed crops, low genetic diversity (neutral or adaptive), or unfavorable genetic correlations due to trade-offs among traits in highly contrasting ecosystems [68].

Our study also indicates how relevant the availability–linked to the existence of previously approved SPA–can be a limiting factor decreasing the potential richness of genetic pools. Despite the important effort made in Europe for approving SPA for many important forest species, not all the regions of provenance are listed in the national registers. This drawback makes often ecosystem restoration activities to be often constrained by a lack of the desired or suitable basic material [33], legal restrictions for seed collection in endangered populations and species [69], and biological constraints such as masting, conservation of seeds and fruits and reduced population size.

The implementation of alternative seed sourcing strategies (*eg.* assisted migration, admixture, climate-adjusted) to maximize the future adaptation and resilience of our forests [19, 66, 70, 71] is not feasible in the short term for most of the species under consideration in this study because we lack precise information about the ecological and evolutionary implications derived from the movement of genetic resources in the landscape [72–74]. Furthermore, many local populations will not be in suitable areas in the future, as in the case of Southern populations of Scots pine [66] or in Maritime pine or Aleppo pine [59], and they do not have any source from southern populations with similar expected future conditions. On one hand, an integrative approach aimed at developing suitable restoration materials for a given area [75] based on their out-planting performance requires well designed field experiments with representative planting stock quality [34, 76]. On the other hand, the rapid development of genomic

research can contribute to optimizing seed sourcing strategies by increasing our knowledge of local adaptation and of the main drivers determining the patterns of variation. The existing genetic characterization by molecular markers and/or field trials (comparative common gardens) is clearly insufficient to improve seed sourcing strategies for most of the species. Obtaining this information is costly in money and time (particularly if it involves robust field experimental settings with multiple sites) since fitness assessment in forest tree species requires long-term experimentation [77].

A growing realization about the importance of SPA for producing reproductive material is the general trend in Europe, where it already represents 84% of the basic materials approved. In Nordic and Central East European countries other basic materials (for producing improved materials) have more importance for some of the studied species, and the election of planting stock has to be based on *improved /cultivar seed sourcing strategies* [26, 78–80] not considered in our study. SPA could provide material for restoration and sustainable forestry [81] and restoration [7]. In order to improve the SPA network, it would be necessary to improve the information on the characteristics of the SPA, as the actual information is mostly limited to the location, and to extend to define deployment zones for the different materials based in strategies aimed to sustainable use and restoration to facilitate the election by users.

## Conclusions

Local seed sourcing strategies should be the basis for sustainable forestry until more scientifically contrasted information is available. Strict-sense local and Wide-sense local seed sourcing strategies provide sufficient species and provenance pools for deployment zones. These pools show a climatic gradient associated with higher annual rainfall and temperatures. The availability of SPA from specific sources reduces the pools for the different deployment zones, despite the huge efforts done in approving SPAs in Spain in particular and the EU in general. Therefore, it will be necessary to increase planning efforts and offer reproductive materials from specific regions to meet demand.

## Supporting information

**S1 Annex. Niche modelling and climatic suitability of the species and genetic pools.**
(DOCX)

**S1 Fig. Deployment regions defined in Spain (from García del Barrio et al. 2001.** *Regiones de identificacion y utilizacion de material forestal de reproduccion*. Serie Cartografica. Madrid: MAPA). BDLJE CC-BY 4.0 ign.es, DR miteco.gob.es.
(DOCX)

**S1 Table. Richness for the seed sourcing pools by deployment zone (DZ).**
(DOCX)

**S2 Table. Richness of seed sourcing pools and deployment zone for 40 forest tree species in Spain.**
(DOCX)

**S3 Table. Data on the use of forest reproductive material of 40 forest tree species in Spain.**
(DOCX)

**S4 Table. T-test for different variables related to the genetic pools and use of forest reproductive material of the species, comparing species with regions of provenance delineated using the divisive *vs*. agglomerative methods.**
(DOCX)

## Acknowledgments

We thank F. J. Auñón and D. Sánchez de Ron (ICIFOR-INIA, CSIC) for their contribution in niche modelling of the different species included in this study. P.C. Grant revised the English version.

## Author Contributions

**Conceptualization:** Ricardo Alía, Eduardo Notivol, José Climent, Juan Majada.

**Data curation:** Diana Barba.

**Formal analysis:** Ricardo Alía, Eduardo Notivol, José Manuel García del Barrio.

**Funding acquisition:** Ricardo Alía, José Climent, Felipe Pérez.

**Validation:** Felipe Pérez.

**Writing – original draft:** Ricardo Alía.

**Writing – review & editing:** Eduardo Notivol, José Climent, Felipe Pérez, Diana Barba, Juan Majada, José Manuel García del Barrio.

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
