## [Decision Letter · Decision Letter 0]

14 Sep 2022

PONE-D-22-22469Local seed sourcing for sustainable forestryPLOS ONE

Dear Dr. Alia,

Thank you for submitting your manuscript to PLOS ONE. After careful consideration, we feel that it has merit but does not fully meet PLOS ONE’s publication criteria as it currently stands. Therefore, we invite you to submit a revised version of the manuscript that addresses the points raised during the review process.

We look forward to receiving your revised manuscript.

Kind regards,

Pankaj Bhardwaj, Ph.D.

Academic Editor

PLOS ONE

Journal Requirements:

"This research was supported by the Project AEG 17-048 established in the frame of the measure 15.2“support to the conservation and use of forest genetic resources” and under Regulation (EU) No 1305/2013 of the European Parliament and of the Council of 17 December 2013 on support for rural development by the European Agricultural Fund for Rural Development (EAFRD) with 75% co-financing. This study was funded by the European Union Horizon 2020 research and innovation programme under grant agreement No 773383 (B4EST project), and the Ministry of Science (RTI2018-094691-B-C32)."

Please state what role the funders took in the study.  If the funders had no role, please state: ""The funders had no role in study design, data collection and analysis, decision to publish, or preparation of the manuscript.""  If this statement is not correct you must amend it as needed. 

"This research was supported by the Project AEG 17-048 established in the frame of the measure 15.2“support to the conservation and use of forest genetic resources” and under Regulation (EU) No 1305/2013 of the European Parliament and of the Council of 17 December 2013 on support for rural development by the European Agricultural Fund for Rural Development (EAFRD) with 75% co-financing. This study was funded by the European Union Horizon 2020 research and innovation programme under grant agreement No 773383 (B4EST project), and the Ministry of Science (RTI2018-094691-B-C32)."

"This research was supported by the Project AEG 17-048 established in the frame of the measure 15.2“support to the conservation and use of forest genetic resources” and under Regulation (EU) No 1305/2013 of the European Parliament and of the Council of 17 December 2013 on support for rural development by the European Agricultural Fund for Rural Development (EAFRD) with 75% co-financing. This study was funded by the European Union Horizon 2020 research and innovation programme under grant agreement No 773383 (B4EST project), and the Ministry of Science (RTI2018-094691-B-C32)."

5. We note that Figures 1 and 3 in your submission contain map images which may be copyrighted. All PLOS content is published under the Creative Commons Attribution License (CC BY 4.0), which means that the manuscript, images, and Supporting Information files will be freely available online, and any third party is permitted to access, download, copy, distribute, and use these materials in any way, even commercially, with proper attribution. For these reasons, we cannot publish previously copyrighted maps or satellite images created using proprietary data, such as Google software (Google Maps, Street View, and Earth). For more information, see our copyright guidelines: http://journals.plos.org/plosone/s/licenses-and-copyright.

a. You may seek permission from the original copyright holder of Figures 1 and 3 to publish the content specifically under the CC BY 4.0 license.  

6. Please upload a new copy of Figure 2 as the detail is not clear. Please follow the link for more information:

https://blogs.plos.org/plos/2019/06/looking-good-tips-for-creating-your-plos-figures-graphics/

https://blogs.plos.org/plos/2019/06/looking-good-tips-for-creating-your-plos-figures-graphics/

Reviewers' comments:

Reviewer's Responses to Questions

**Comments to the Author**

1. Is the manuscript technically sound, and do the data support the conclusions?

Reviewer #1: Yes

Reviewer #2: Yes

Reviewer #3: Yes

2. Has the statistical analysis been performed appropriately and rigorously? 

Reviewer #1: No

Reviewer #2: Yes

Reviewer #3: Yes

3. Have the authors made all data underlying the findings in their manuscript fully available?

Reviewer #1: Yes

Reviewer #2: Yes

Reviewer #3: No

4. Is the manuscript presented in an intelligible fashion and written in standard English?

Reviewer #1: Yes

Reviewer #2: Yes

Reviewer #3: Yes

5. Review Comments to the Author

Reviewer #1: This work analyzed two seed sources strategies (strict-sense vs wide-sense local seed sources) for 40 species and 50 deployment sites in Spain and in relation to the richness of the pools, their climatic trends and their availability of basic material. The authors highlight the importance of these two strategies as they can sustain good levels of species and genetic background, while also discussing some limitations. The work is interesting as it deals with technical and scientific aspects relevant for sustainable forestry. However, there are some aspects that need to be addressed, and are related to make the work clearer for the reader. For instance, very specific terminology "strict-sense, wide-sence, basic materials, selected-material, etc, need to be defined as the audience of PlosONE is of broder scope, also some of the mentioned regulations may not be clear for scientists outside the EU. Another aspect is a better justification/explanation of the statistical methods. Specific comments can be found below:

Ln26.27: I don’t understand what this means

The abstract is not clear, I don´t understand which is the type of data that has been analyzed (ln27-29) and which were the specific variables tested (ln30), and neither which were the results (ln33). The main aim refers to limitations for the application of local seed sourcing, but which limitations? Another ambiguity is what the authors refer to “actual regulations on marketing”? For reader that is not familiar with the market in EU countries, this conclusion has no context. Which are the actual regulations? If I read this paper in 2 - 5 years ahead, or more, this line would have no significance. So please make the abstract easier to understand, present specific results (report some values, statistical test performed, significant differences, etc.) and provide a conclusion of a broader view.

Ln51: Move this sentence in the same first paragraph. Also the sentence is long and is difficult to read. Improve writing.

Ln80: Put in parenthesis, 1 or 2 examples of the existing regulations, so it is easier to understand which would be a limitation; because I don’t understand the “like ecosystem restoration”

Ln82-113: The first part of the introduction is clear as it defines some terminology, which is broader in context. However, the last two paragraphs that explains the particularities of this work and the aims are not that clear to a reader that has no specific context of the terminology or regulations in Spain or EU. If that would be the case, then the work is suitable to a more specific journal, but it can be resolved if more context, and terminology definition is provided. Also check if the terminology is the most used. For e.g. source-identified and selected reproductive materials in the literature are named as identified stands and selected stands.

Ln82: mention the difference between source-identified and selected reproductive materials. As I mentioned earlier, its important to define some terminology to target a wider audience as it’s the aim of PlosONE.

Ln86: What refers to “low selection of the material”?

Ln105: Specify which two genetic levels, the writing is a bit confusing

Ln136-137: can you refer to a map (supplementary material)?

Ln144: forest species 17 species.. correct

Ln148: SSL and WSL abreviattions should be presented since the first time both terms are presented

Ln173: What means FRM?

Ln196-199: Be more specific about the statistical model applied, the explanation is very vague. In which software? Also does the variables were standardized/transformed? How does the model fit was checked (for the regression and t-test)? Which was the alpha level?

Also From where the climatic variables were obtained?

Ln241: Move this sentence with previous one

Ln278: countries

The figures are not clear. For instance fig 2, the coloring (red vs blue) and the IDs?

Fig 3. Not sure what the figures depicts. The category numbers and state again the abbreviation significance (MA_is).

Ln346: Therefore.. incomplete sentence

I feel that the discussion needs to expand more on the specific results found in this study, for instance expand on the differences found between SSL and WSL.

Reviewer #2: The manuscript has been prepared well. Few suggestions need to be considered, to begin with, you may add other appealing charts rather than tables with the help of many standard software's. Additionally, modify the materials and methods section by mentioning the duration of the experiment and lastly check grammatical /typo error mistakes.

Reviewer #3: The study is an overview of the current status of the available genetic resources of forest reproductible material in regards to their potential use. The authors elaborate and adhoc methodology applied to Spain, and then further extended it to Europe (sensu European Union). To my knowldege , this is the first overall analysis of this issue conducted at the scale of a country and a continent. The issue is timely in the context of the continuous extension of forest plantation and afforestation driven by the need of climate warming mitigation. Appropriate seed sourcing is of utmost importance in this context. I would therefore strongly recommend that the paper be published.

However the current version has some limitations, that incline me to exhort the authors to provide an improved version before final publication. The current manuscript is extremely concise and can only be understood by scientists and professionals familiar to operational forestry. To be accessible to a larger audience, clarifications and additional information is needed at various places in the manuscript. I suggest below items and places where additional clarifications would be relevant.

Concerning the analysis per se, it appears to me that the outcomes of the study tighty depend on the way the procurement zones and the deployment zones are geographically distributed and defined. While the delineation of the procurement zones is actually imposed by the regulation scheme in place (EU or SP), the deployment zone can be envisaged according to different reasonings. The authors decided to use a subdivision of Spain « in continuous regions with similar environmental characteristics ». The authors should first clarify what « similar environmental characteristics » stand for. And second, they should consider how sensitive the method is if « different environmemntal charateristics » are used. What if Spain is subdivided in only 30 units ? or 70 units ? Alternatively they should provide the rationale for using only 50 units.

The same limitations apply to the use of niche modelling. They are just multiple ways of implementing niche modelling. And they may end up in different outcomes of the study. Instead of building a theoretical distribution of the species based on niche modelling, why not just consider the today’s distribution ? This would be less sensitive to the choice of climatic variables used for niche modelling.

Line 82-83 It would be relevant to define source-identified and selected material.

Are source identified stands phenotypcally selected stands ? Were they identified based on dendrological, phytosanitary or other criteria ?

Line 86-87 Where is the remaining material for deployment coming from (besides source-identifed and selected) that represent 79.2% ?

Line 94-95 Do you mean « between 82 to 85% »… and « between 14 to 17% »

Line 107-108 What do you mean by « Climatic trends on the richness of these pools »

Line 108-109 What do you mean by « the relationship among richness of the pools, the use of reproductive material of the species and the method for defining regions of provenance »

Table 1 : Why are there 2 different regulation schemes : SP and EU ? Why do all species not undergo the EU scheme ? Is it because they are absent in the EU scheme ?

Line 145 Explain what a divisive method is ? Clarify how the regions of provenances were » assembled » according to the divisive method. Or provide a reference that would clarify the procedure.

Line 152-156 What climatic variables were used to calculate the matrix of the Mahalanobis distances ? The whole method can also be sensitive to the type of climatic information used.

Line 189-190 Briefly explain what the FOREMATIS data base contains ??

Line 196-205 Were the climatic variables averaged for each deployment zone ? How were the raw climatic data estimated at the level of a deployment zone ?

Line 213-216 Unclear. Could you rephrase or elaborate more on the computation used.

Line 219-220 Redundancy with line 208-209

Line 330-334 Doesn’t the wide sense local seed-sourcing strategy fit into the assisted-migration approach ? And if so, your results suggest that assisted migration is feasible. So I do not understand here why assisted migation or climate-adjusted migration is not suitable.

Line 346. Remove « therefore ». Unless this is the start of a sentence that is missing. If so, add the sentence.

Line 348 What does « 84 % » correspond to ??

Line 350 substitue « based in « to « based on ».

Line 356-357 You may elaborate more on how you results may stimulate to « homogenize regulations concerning the use of Forest Reproductive material »

Figure 1 I was unable to read the numbers on the different maps. Please provide pictures with higher resolution.

Figure2 same comment as for Figure 2

Figure 3 Homogenizethe writings of acronyms between Figure 3 and Table 3

6. PLOS authors have the option to publish the peer review history of their article (what does this mean?). If published, this will include your full peer review and any attached files.

Reviewer #1: No

Reviewer #2: **Yes: **Dr. Somdutt Sharma

Reviewer #3: No

---

## [Author Response · Author response to Decision Letter 0]

7 Nov 2022

Response to reviewers

Dear Editor,

Many thanks for the review of our paper on local seed sourcing for sustainable forestry. We have carefully read the comments, and we have tried to improve our manuscript accordingly.

Concerning the Journal requirements, there are several comments.

- Financial PLOS ONE's style requirements. We have checked the files format.

- Financial disclosure. The funders had no role in study design, data collection and analysis, decision to publish, or preparation of the manuscript. We have included this text after the funding information.

- Acknowledgements. We have removed any funding-related text from the manuscript (sorry for the mistake in the previous version), and we have included the contribution of different persons involved in the preliminary research.

- Financial statement: Here we include the correct statement.

This research was supported by the Project AEG 17-048 established in the frame of the measure 15.2“support to the conservation and use of forest genetic resources” and under Regulation (EU) No 1305/2013 of the European Parliament and of the Council of 17 December 2013 on support for rural development by the European Agricultural Fund for Rural Development (EAFRD) with 75% co-financing (FP). This study was funded by the European Union Horizon 2020 research and innovation programme under grant agreement No 773383 (B4EST project) (RA), and the Ministry of Science (RTI2018-094691-B-C32) (JC).

- Figures permission: Figures 1, 3, S1. We used public information for the maps from the Spanish National Institute of Geography (IGN) under license CC-BY 4.0 (https://www.ign.es/resources/licencia/Condiciones_licenciaUso_IGN.pdf) and for the Deployment Regions (DR) the Ministry of Ecological transition (MITECO) (Public information as stated by the Resolution of the Ministry on regions of provenance of forest tree species, BOE 224, 16/09/2009). For Figure 3, we use information from www.forematis.eu, on a political map of Europe (https://es.m.wikipedia.org/wiki/Archivo:Political_Map_of_Europe-en.svg). CC BY 4.0. 

- New Figure 2. We have included a new figure 2, following Plos One policy on figure creation. 

- Captions in SI. We have also checked and completed the captions in SI.

We addressed all the comments by the reviewers, as follows:

Reviewer #1: This work analyzed two seed sources strategies (strict-sense vs wide-sense local seed sources) for 40 species and 50 deployment sites in Spain and in relation to the richness of the pools, their climatic trends and their availability of basic material. The authors highlight the importance of these two strategies as they can sustain good levels of species and genetic background, while also discussing some limitations. The work is interesting as it deals with technical and scientific aspects relevant for sustainable forestry. However, there are some aspects that need to be addressed, and are related to make the work clearer for the reader. For instance, very specific terminology "strict-sense, wide-sence, basic materials, selected-material, etc, need to be defined as the audience of PlosONE is of broder scope, also some of the mentioned regulations may not be clear for scientists outside the EU. Another aspect is a better justification/explanation of the statistical methods. Specific comments can be found below:

Thanks for your comment. We have modified extensively the introduction, methods and discussion. We have clarified the terminology by including more detailed descriptions. You can check in the 'Revised Manuscript with Track Changes' file all the changes. Also, we have clarified the connexion among the EU regulation with other related ones (eg. OCDE, some USA states) and in connexion with the usual terminology in other papers on this topic.

Ln26.27: I don’t understand what this means.

The abstract is not clear, I don´t understand which is the type of data that has been analyzed (ln27-29) and which were the specific variables tested (ln30), and neither which were the results (ln33). The main aim refers to limitations for the application of local seed sourcing, but which limitations? Another ambiguity is what the authors refer to “actual regulations on marketing”? For reader that is not familiar with the market in EU countries, this conclusion has no context. Which are the actual regulations? If I read this paper in 2 - 5 years ahead, or more, this line would have no significance. So please make the abstract easier to understand, present specific results (report some values, statistical test performed, significant differences, etc.) and provide a conclusion of a broader view.

We have changed the abstract to clarify the questions raised by the reviewer. One of the major limitation (as stated in the introduction line.) is that the reproductive material of the desired origin or seed source is often not available in the market (seed companies, nurseries) . This is especially true when considering species or origins of low commercial value or with low interest in terms of seed or plant supply.

Also, the regulations establish, for forestry purposes, which material you can collect and from where, as they have to be officially approved. We have tried to clarify these two important topics in the in the introduction and discussion, in order to understand why we use an available seed pool based on the existing basic material approved for a given species.

Ln51: Move this sentence in the same first paragraph. Also the sentence is long and is difficult to read. Improve writing.

Thanks for your suggestion. We have included the sentence in the first paragraph and we have rewritten the sentence.

Ln80: Put in parenthesis, 1 or 2 examples of the existing regulations, so it is easier to understand which would be a limitation; because I don’t understand the “like ecosystem restoration”.

We have rewritten this paragraph to clarify the different regulations (similar among different regions /countries, with some differences in terminology) and the basis of those regulations. The main bottleneck, particularly for ecosystem restoration, is that using strict-sense local material, implies collecting in the same area you are deploying the material. Therefore, you will need an extensive network of seed collection areas for a given species that is usually commercially unfeasible. The references are related to these limitations.

Ln82-113: The first part of the introduction is clear as it defines some terminology, which is broader in context. However, the last two paragraphs that explains the particularities of this work and the aims are not that clear to a reader that has no specific context of the terminology or regulations in Spain or EU. If that would be the case, then the work is suitable to a more specific journal, but it can be resolved if more context, and terminology definition is provided. Also check if the terminology is the most used. For e.g. source-identified and selected reproductive materials in the literature are named as identified stands and selected stands.

We have rewritten these paragraphs to help understanding the framework of seed sourcing of species under regulation for seed trading, in the context of sustainable forest management. We have also explained some specific terminology, especially the relationship between basic and reproductive materials, their requisites and categories. In particular, identified stands (seed sources in the EU) will provide seed of the identified category, and selected stands will provide selected category seed. We have also decide to use the concept of Seed Production Area (ASP), a more general one that can be applied both in afforestation and restoration context. This way, identified or selected seed stands can be considered particular cases of ASPs.

Ln82: mention the difference between source-identified and selected reproductive materials. As I mentioned earlier, its important to define some terminology to target a wider audience as it’s the aim of PlosONE.

As mentioned before, we have entirely rewritten this part to clarify the difference, and why they are important in the context of sustainable forestry.

Ln86: What refers to “low selection of the material”?

We have explained in detail the type of selection for the different categories of seeds/plants. 

Ln105: Specify which two genetic levels, the writing is a bit confusing.

We have rephrased the complete paragraph. The levels (species or region of provenance) are explicitly cited when appropriate.

Ln136-137: can you refer to a map (supplementary material)?

We have included a Figure S1 with this information.

Ln144: forest species 17 species.. correct

Thanks. Done.

Ln148: SSL and WSL abreviattions should be presented since the first time both terms are presented

Thanks, we have corrected the paragraph.

Ln173: What means FRM?

We have eliminated the acronym throughout the manuscript.

Ln196-199: Be more specific about the statistical model applied, the explanation is very vague. In which software? Also does the variables were standardized/transformed? How does the model fit was checked (for the regression and t-test)? Which was the alpha level? Also From where the climatic variables were obtained?

We have improved the description of the methods used, with the software used (R packages), how we obtained the climatic variables, the computation of mean values by deployment region and the standardization and significance levels.

Ln241: Move this sentence with previous one

Done.

Ln278: countries

Done.

The figures are not clear. For instance fig 2, the coloring (red vs blue) and the IDs?

Fig 3. Not sure what the figures depicts. The category numbers and state again the abbreviation significance (MA_is).

We have improved Figure 2, and 3, included a Figure 4 for the ANOVA results and revised all captions of the different figures.

Ln346: Therefore.. incomplete sentence

Thanks. We have corrected the sentence.

I feel that the discussion needs to expand more on the specific results found in this study, for instance expand on the differences found between SSL and WSL.

Thanks. We have focused in our specific results, and especially in the differences between SSL and WSL strategies

Reviewer #2: The manuscript has been prepared well. Few suggestions need to be considered, to begin with, you may add other appealing charts rather than tables with the help of many standard software's. Additionally, modify the materials and methods section by mentioning the duration of the experiment and lastly check grammatical /typo error mistakes.

Thanks for your review. We have improved the figures for the clarification of the results, especially Figure 2 by increasing the font size, changing the colors, and improving the legend. Also, we have improved Figure 1 and 3, deleting non-relevant information (as the code of deployment regions, and information on the categories of basic material. Considering the addition of new charts, rather than tables, we have included the figure 4 with information from Table 3 (deleted in this new version) to facilitate the reading and visualization of the results.

Reviewer #3: The study is an overview of the current status of the available genetic resources of forest reproductible material in regards to their potential use. The authors elaborate and adhoc methodology applied to Spain, and then further extended it to Europe (sensu European Union). To my knowldege , this is the first overall analysis of this issue conducted at the scale of a country and a continent. The issue is timely in the context of the continuous extension of forest plantation and afforestation driven by the need of climate warming mitigation. Appropriate seed sourcing is of utmost importance in this context. I would therefore strongly recommend that the paper be published.

Thanks for your comment. 

However the current version has some limitations, that incline me to exhort the authors to provide an improved version before final publication. The current manuscript is extremely concise and can only be understood by scientists and professionals familiar to operational forestry. To be accessible to a larger audience, clarifications and additional information is needed at various places in the manuscript. I suggest below items and places where additional clarifications would be relevant.

We agree that the papers should be aimed to a broader audience. We have improved the descriptions following your recommendations. 

Concerning the analysis per se, it appears to me that the outcomes of the study tighty depend on the way the procurement zones and the deployment zones are geographically distributed and defined. While the delineation of the procurement zones is actually imposed by the regulation scheme in place (EU or SP), the deployment zone can be envisaged according to different reasonings. The authors decided to use a subdivision of Spain « in continuous regions with similar environmental characteristics ». The authors should first clarify what « similar environmental characteristics » stand for. And second, they should consider how sensitive the method is if « different environmemntal charateristics » are used. What if Spain is subdivided in only 30 units ? or 70 units ? Alternatively they should provide the rationale for using only 50 units.

This is an important topic, and we have tried to clarify in the text. Different approaches use methods based on population characteristics. However, we think that is more adequate to use a more suitable scale for the use of reproductive material. Regions of provenance (or seed zones) and deployment regions are of similar scale, and they are both related to the scale at which the forest tree species are genetically different or can respond similarly to environmental conditions. Therefore we think that our results are not biased by the number of regions considered, unless these regions were not appropriately defined. We have discussed all these aspects to justify the validity of our approach (lines 303-314). 

The same limitations apply to the use of niche modelling. They are just multiple ways of implementing niche modelling. And they may end up in different outcomes of the study. Instead of building a theoretical distribution of the species based on niche modelling, why not just consider the today’s distribution ? This would be less sensitive to the choice of climatic variables used for niche modelling.

We have clarified this point. We use actual distribution of the species for defining the local material, and we use niche modelling to define the suitable area of each species, and based on this area, define the wide-sense local material. 

Line 82-83 It would be relevant to define source-identified and selected material.

Are source identified stands phenotypcally selected stands? Were they identified based on dendrological, phytosanitary or other criteria?

We have included the main characteristics of the source-identified and selected material. In the case of source-identified, the origin is the criteria for approval, and in the case of selected stands, a phenotypic selection at the population (stand) level. 

Line 86-87 Where is the remaining material for deployment coming from (besides source-identifed and selected) that represent 79.2% ?

We have clarified that the remaining are coming from qualified and tested materials.

Line 94-95 Do you mean « between 82 to 85% »… and « between 14 to 17% »

Sorry, there was an error from an early draft version distinguishing between seeds and plants. Now we used only one figure for each case.

Line 107-108 What do you mean by « Climatic trends on the richness of these pools »

We mean that richness is related to different climatic variables. We have rephrased the sentence.

Line 108-109 What do you mean by « the relationship among richness of the pools, the use of reproductive material of the species and the method for defining regions of provenance »

We have rephrased the text.

Table 1 : Why are there 2 different regulation schemes : SP and EU ? Why do all species not undergo the EU scheme ? Is it because they are absent in the EU scheme?

Some species are regulated in Spain, but not at the European level. However, as the two schemes have the same principles, we have deleted this distinction.

Line 145 Explain what a divisive method is ? Clarify how the regions of provenances were » assembled » according to the divisive method. Or provide a reference that would clarify the procedure.

We have rewritten this part to clarify the differences, with relevant references.

Line 152-156 What climatic variables were used to calculate the matrix of the Mahalanobis distances ? The whole method can also be sensitive to the type of climatic information used.

The variables have been included in the main text. These variables are the most important for describing the distribution of the species, and therefore we think are good indicators to measure the climatic distance among sites (Annex S1). For the computation of the Mahalanobis the distance, we are using a complete set of variables, and therefore we do not expect biases due to the variables used in our analysis.

Line 189-190 Briefly explain what the FOREMATIS data base contains??

We have included a brief description.

Line 196-205 Were the climatic variables averaged for each deployment zone? How were the raw climatic data estimated at the level of a deployment zone?

The climatic variables were computed for one km2 grid, and then averaged for the different regions. It has been clarified.

Line 213-216 Unclear. Could you rephrase or elaborate more on the computation used.

We have included the method used.

Line 219-220 Redundancy with line 208-209

We deleted in this part, and improved the description in the lines 208-209. Thanks. 

Line 330-334 Doesn’t the wide sense local seed-sourcing strategy fit into the assisted-migration approach? And if so, your results suggest that assisted migration is feasible. So I do not understand here why assisted migation or climate-adjusted migration is not suitable.

We have rephrased this sentence, as we refer to assisted migration from southern populations as in many occasions there are no such kind of populations in the north of Africa.

Line 346. Remove « therefore ». Unless this is the start of a sentence that is missing. If so, add the sentence.

Done. Thanks

Line 348 What does « 84 % » correspond to ??

We have included that source-identified basic material represent 84% all the approved in the EU.

Line 350 substitue « based in « to « based on ».

Done. Thanks

Line 356-357 You may elaborate more on how you results may stimulate to « homogenize regulations concerning the use of Forest Reproductive material »

We have rephrased this sentence. 

Figure 1 I was unable to read the numbers on the different maps. Please provide pictures with higher resolution. Figure2 same comment as for Figure 2. Figure 3 Homogenizethe writings of acronyms between Figure 3 and Table 3

We have improved the quality of the figures and the captions. We have included a new Figure S1 with the codes of the Deployment regions, and a new Figure 4 with the comparison among EU regions.

---

## [Decision Letter · Decision Letter 1]

28 Nov 2022

Local seed sourcing for sustainable forestry

PONE-D-22-22469R1

Dear Dr. Alia,

We’re pleased to inform you that your manuscript has been judged scientifically suitable for publication and will be formally accepted for publication once it meets all outstanding technical requirements.

Kind regards,

Pankaj Bhardwaj, Ph.D.

Academic Editor

PLOS ONE

Additional Editor Comments (optional):

Reviewers' comments:

Reviewer's Responses to Questions

**Comments to the Author**

1. If the authors have adequately addressed your comments raised in a previous round of review and you feel that this manuscript is now acceptable for publication, you may indicate that here to bypass the “Comments to the Author” section, enter your conflict of interest statement in the “Confidential to Editor” section, and submit your "Accept" recommendation.

Reviewer #1: All comments have been addressed

Reviewer #2: All comments have been addressed

Reviewer #3: All comments have been addressed

2. Is the manuscript technically sound, and do the data support the conclusions?

Reviewer #1: Yes

Reviewer #2: Yes

Reviewer #3: Yes

3. Has the statistical analysis been performed appropriately and rigorously? 

Reviewer #1: Yes

Reviewer #2: Yes

Reviewer #3: Yes

4. Have the authors made all data underlying the findings in their manuscript fully available?

Reviewer #1: Yes

Reviewer #2: Yes

Reviewer #3: Yes

5. Is the manuscript presented in an intelligible fashion and written in standard English?

Reviewer #1: Yes

Reviewer #2: Yes

Reviewer #3: Yes

6. Review Comments to the Author

Reviewer #1: I have read this manuscript's revised version and consider it improved significantly. I just have a few minor editorial remarks:

Ln53: Put as Seed Production Area

Ln56: remove both

Ln107: said richness?

Ln111: Such as the Mediterranean forests

Ln127: remove “according”

Ln149: variables

Ln158: Add a comma after similarity

Ln306: correct the sentence “These regions are in the”

Ln311: Separate genepools

Ln340: remove “as”

Reviewer #2: The Author has modified the manuscript as per all the suggestions with new figures. You may change the the view of new figures and by modifying its zoom for clarity.

Reviewer #3: You have addressed all my remarks concerns by doing relevant corrections and providing convincing arguments

7. PLOS authors have the option to publish the peer review history of their article (what does this mean?). If published, this will include your full peer review and any attached files.

Reviewer #1: No

Reviewer #2: **Yes: **Dr. Som Dutt Sharma

Reviewer #3: No

---

## [Editor Report · Acceptance letter]

2 Dec 2022

PONE-D-22-22469R1 

Local seed sourcing for sustainable forestry 

Dear Dr. Alía:

I'm pleased to inform you that your manuscript has been deemed suitable for publication in PLOS ONE. Congratulations! Your manuscript is now with our production department. 

Kind regards, 

on behalf of

Dr. Pankaj Bhardwaj 

Academic Editor

PLOS ONE